# Chemical Composition, Antioxidant Properties, and Antibacterial Activity of Essential Oils of *Satureja macrostema* (Moc. and Sessé ex Benth.) Briq

**DOI:** 10.3390/molecules28124719

**Published:** 2023-06-12

**Authors:** Lucia Barrientos Ramírez, José Antonio Silva Guzmán, Edison Antonio Osorio Muñoz, Carlos Alvarez Moya, Mónica Reynoso Silva, Abraham Francisco Cetina Corona, Josefina Casas Solis, J. Jesús Vargas Radillo

**Affiliations:** 1Department of Wood, Cellulose and Paper, University of Guadalajara, CUCEI, Road Ing. Ramón Padilla Sánchez 2100, Las Agujas, Zapopan 45200, Jalisco, Mexico; 2Group of Research and Development in Science Applied to Biological Resources, Salesian Polytechnic University, 12 de Octubre Avenue N2422 and Wilson, Quito 170109, Ecuador; 3Environmental Mutagenesis Laboratory, Cellular and Molecular Department, University of Guadalajara, Guadalajara 45200, Jalisco, Mexico; 4Microbiology Laboratory, Cellular and Molecular Department, University Center for Biological and Agricultural Sciences (CUCBA), Road Ing. Ramón Padilla Sánchez 2100, Nextipac, Zapopan 45200, Jalisco, Mexico

**Keywords:** piperitone derivatives, direct bioautography, TEAC, Lamiaceae, chemotypes

## Abstract

*Satureja macrostema* is a plant that is located in various regions of Mexico and is used in a traditional way against illness. Essential oils (EOs) were obtained from leaves *Satureja macrostema* and the chemical composition was evaluated by gas chromatography–mass spectrometry (GC-MS). The antioxidant effect of the oil was assayed by 2,2-diphenyl-1-picrylhydrazyl (DPPH) and by Trolox Equivalent Antioxidant Capacity (TEAC). In vitro antibacterial activity against *Escherichia coli* and *Staphylococcus aureus* was determined using a broth microdilution assay and thin layer chromatography–direct bioautography (TLC-DB) to identify active antibacterial compounds. The EOs analysis showed 21 compounds, 99% terpenes, and 96% oxygenated monoterpenes, with *trans*-piperitone epoxide (46%), *cis*-piperitone epoxide (22%), and piperitenone oxide (11%) as more abundant compounds. Likewise, *S. macrostema* EOs showed an antioxidant activity of DPPH = 82%, with 50% free radical scavenging (IC_50_) = 7 mg/mL and TEAC = 0.005, an antibacterial effect against *E. coli* of 73% inhibition, and 81% over *S. aureus* at dose of 100 µL of undiluted crude oil. The TLC-DB assay showed that the most active compounds were derived from piperitone. The comparison with other studies on *S. macrostema* shows variability in the compounds and their abundances, which can be attributed to climatic factors and the maturity of plants with similar antioxidant and antibacterial activities.

## 1. Introduction

The genus *Satureja* (Family Lamiaceae, tribe Mentheae) contains about 200 species, with a presence in the midwest of the European Mediterranean, West Asia, North Africa, and South America [1]. *Satureja macrostema* (Moc. and Sessé ex Benth.) Briq. (Lamiaceae), commonly called nurite or nurhitini (purepecha from Michoacan), poleo, tea bush, or tuché (Mixe from Oaxaca state), has been used since pre-Hispanic times in traditional herbalism. It is a Mexican medicinal plant of great importance in various mountain areas with temperate or semi-cold climates in the western part of the country, such as Michoacán, Oaxaca, Jalisco, and San Luis Potosi [2]. The region includes mountainous, volcanic, and forested areas of high biodiversity and floristic composition, with various plants with nutritional or medicinal properties occurring in the so-called “purepecha plateau” in Michoacan or in the central valleys of Oaxaca, where *S. macrostema* grows naturally and is discontinuously distributed below the main plant canopy (pine and oak forests) as part of the undergrowth [3]. It is a shrub with the scent of mint when squeezed; it stands 1 to 2 m high with erect stems and arched branches. The leaves are 1–4 cm long by 0.6–1.5 cm wide, with sharp apices and serrated and rounded bases. The flowers are solitary, corolla red or orange, and 2–3.5 cm length. The plant flowers from July to October and bears fruit from September to November [2]. This species is also known by the synonyms *Melissa macrostema* Moc. and Sessé ex Benth., *Calamintha macrostema* (Moc. and Sessé ex Benth.) Benth., and *Clinopodium macrostemun* (Moc. and Sessé ex Benth.) [4]. The leaves, flowers, and stems, and especially the dried leaves, of *S. macrostema* are used in traditional medicine as an infusion, called nurité tea, which is said to be effective against muscle aches, nausea, diarrhea, infectious diseases, and even against infertility and hangovers [5]. Phytochemical studies have revealed the presence of phenolic compounds, sterols, and volatile essential oils such as carvacrol and thymol in *Satureja* [6]. A characteristic of this genus of aromatic plants is their content of essential oils (EOs) (>0.5%). Terpenes are the main component, as in medicinal plants, and have antioxidant properties [7], but with different chemical compositions among the subspecies [8]. There are studies in both extracts and essential oils of the various species of *Satureja* which evaluate, above all, the antioxidant activity [9,10]. Likewise, the EOs are affected by extraction methods such as conventional hydrodistillation (HD), steam distillation (SD), and solvent extraction, as well as the modern techniques of supercritical carbon dioxide (scCO_2_) and microwave-assisted extraction (MAE). The commonly used techniques are HD, SD, and MAE due to costs, system simplicity, and extraction durations, but these have the disadvantages of low yield, low quality, and high temperature [11]. To obtain the EOs in this work, we used HD due to its simplicity and frequent use, while recognizing the advantages of the most efficient modern techniques. There are few studies on *S. macrostema* essential oils (*S. macrostema* EOs) and their biological activity, so the objective of this study was to determine the chemical composition of *S. macrostema* collected in the wild under environmental conditions on the Purepecha plateau. We also evaluated the effect on *E. coli* and *S. aureus*, determining the chemical compounds responsible for this microbicidal activity. 

## 2. Results

### 2.1. Chemical Characterization of Essential Oils

A total of 21 compounds were detected in the EOs according to the essential oil MS library, and these are presented in Table 1. The table consists of eight columns, and the abbreviations are described in the footer of the table. The seventh column (% relative area) indicates the relative concentration of the compounds. The last column (S.D.) consists of the standard deviation of three replicates. The components are listed in order of elution. The nomenclature is in accordance with NIST (National Institute of Standards and Technology) [12]. The identification methods of the compounds were MS library matching [12,13] and linear retention indices (LRI) [13]. The identified compounds were also compared for their coincidence with the IR of bibliographic reports [14,15,16]. The percentage of identified compounds was 99% (1% unidentified). The compounds with the highest presence were the oxygenated monoterpenes derived from *trans*-piperitone epoxide ketones (C_10_H_16_O_2_) (46%), *cis*-piperitone epoxide (22%), piperitone oxide (11%), 3-octanol acetate (6%), linalool (5%), pulegone (3%), and menthone (3%).

### 2.2. DPPH and TEAC Free Radical Scavenging Assay

Table 2 shows the antioxidant activity of *S. macrostema* essential oils, with limonene as a control (since it is a component of essential oils with antioxidant activity) and Trolox as a reference. Limonene (an antioxidant monoterpene) and Trolox (6-hydroxy-2,5,7,8-tetramethylchroman-2-carboxylic acid, a compound with strong antioxidant activity) are included as references. In addition, the data of Table 2 were plotted and shown in Figure 1 before the zone of 50% inhibition was located. The data from Table 2 were plotted as shown in Figure 1a,c,d. Next in Figure 1 graphs a and b, the straightest area in which 50% of the initialization is found (orange box) was located to replot this area as shown in Figure 1b,d and then obtain the equation of the line y the half-maximal inhibitory concentration (IC_50_). These were as follows: IC_50_
*S. macrostema* EOs (y = 6.4985x + 2.7954, R^2^ = 0.9932) = 7.3 mg/mL; IC_50_ limonene (y = 0.0815x − 0.0614, R^2^ = 0.9724) = 614 mg/mL; IC_50_ Trolox (y = 1359.2x + 2.3029, R^2^ = 0.98) = 0.03 mg/mL. The DPPH (2,2-diphenyl-1-picryl-hydrazyl-hydrate) value was expressed in Trolox equivalents (TEAC = IC_50_Trolox (mg/mL)/IC50 sample (mg/mL)): TEAC *S. macrostema* EOs = 0.005 y TEAC limonene = 0.00006. High TEAC (Trolox Equivalent Antioxidant Capacity) values signify higher DPPH free radical scavenging or scavenging activity. 

### 2.3. Antimicrobial Activity In Vitro

The in vitro antibacterial activity data of the essential oils of *S. macrostema* at five different dilution levels (undiluted or direct up to 1:1000) are presented in Table 3. During the tests, 100 µL of each dilution of *S. macrostema* EOs was applied over *E. coli* and *S. aureus,* and the level of inhibition of bacterial growth was measured. The greatest effect was obtained by applying the essential oils directly, undiluted, with 73% inhibition on *E. coli* and 81% on *S. aureus*. Observing the data in the Table 3 shows a greater antimicrobial effect on *S. aureus* than on *E. coli.*

### 2.4. Thin Layer Chromatography–Direct Bioautography (TLC-DB)

Figure 2 shows the results of the activity of the EOs from *S. macrostema* (expressed as Sat in the chromatographic plate) on *E. coli* and on *S. aureus*. Two different techniques revealed that double verification of the antimicrobial zones on the TLC plates was required. In the TLC-DB trial (Figure 2) EOs from the plants *Magnolia pugana* (Mp) (Magnoliaceae) and *Thymus vulgaris* (Tv) (Lamiaceae) were also tested to compare the antimicrobial effect of these plants with medicinal properties with the *S. macrostema* EOs. The antibacterial effect manifested itself in the form of light or dark spots, and a larger surface area was interpreted as indicating a greater intensity of the antibacterial effect. In Figure 2, it can be appreciated that the f1, f2, and f3 fractions represented the spots with the largest areas on the chromatographic plate, and, thus, were analyzed by HPLC-MS to evaluate the compounds that caused the antibacterial activity. The results are presented in Table 4, where the major inhibition compounds (>) found were piperitone oxide, *trans*-piperitone epoxide, and linalool, in addition to other compounds such as piperitenone, *cis*-piperitone epoxide, *trans*-thujene, benzaldehyde, caryophyllene, spathulenol, *iso*-menthone, and 2-phenyl ethyl acetate. 

## 3. Discussion

The chemical composition, antioxidant effects, and in vitro antibacterial activity of essential oils from *Satureja macrostema*, an aromatic plant commonly used in some regions of Mexico for the treatment of various diseases, were evaluated. The essential oils of the *S. macrostema* leaves were isolated by hydrodistillation, with a dry basis yield of 0.8% (*w*/*w*). Table 1 shows the chemical composition of *S. macrostema* Eos, which include various volatile oils, such as monoterpene hydrocarbons, oxygenated monoterpenes, aldehydes, ketones, alcohols, phenols, epoxy compounds, oxides, sesquiterpenes (some with roles as non-steroidal anti-inflammatory drugs), oxygenated sesquiterpenes, and oxygenated hydrocarbons. A comparison of these results with those of other studies of *S. macrostema* from the same region (the Purepecha plateau of Michoacan, Mexico) shows differences in the amounts and types of compounds found in essential oils (EOs). The essential oils obtained from the aerial parts (leaves and stems) of *S. macrostema* a (Moc. And Sessé ex Benth.) Briq. Plants, which were attained by micropropagation from seeds collected in experimental plantations in Parangaricutiro Michoacan (19°25′23″ N, 02°07′47″ W), have the major compounds pulegone (25.5%), linalool (16.62%), thymol (14.64%), limonene (5.53%), caryophyllene (3.98%), and menthone (3.09%) [13]. This highlights the absence of piperitone compounds. On the other hand, in another study, the leaves of *S. macrostema* plants from an experimental field in Uruapan Michoacan were treated with a process of prolonged cold maceration and subsequent procurement of the terpene fractions of the essential oils during plant development (15, 30, 45 and 90 days). This showed variation in the presence of some compounds according to the maturity of the plant, with the most significant effect being a decrease in pulegone and an increase in limonene and piperitone. In the highest state of maturity, these plants contain mainly limonene (35.54%), piperitone oxide (25.71%), and piperitone (10.10%), and in lower concentrations, they contain menthol, pulegone, and menthone, among others [17]. In another study, EOs obtained from dry leaves of *Clinopodium macrostemum* (synonym of *S. macrostema*) were collected from the forest of the Mixe region of Oaxaca. They had a yield of 0.8% (*w*/*v*), with 32 compounds identified, among them being linalool (55.4%), caryophyllene (6.3%), menthone (5.8%), geraniol acetate (4.1%), terpineol (3.7%), and pulegone (2.8%) [18]. This variation in plant compounds from the same area of study is not strange or rare and has been documented in other investigations. This was the case of the study carried out on EOs from *Satureja parvifolia* leaves in five populations of Tucuman, Argentina, which showed variability in their chemical compositions, particularly with the presence of pulegone, carvacrol, and piperitone in samples from one region and their absence in samples from another region of the study area [19]. Another study [20] concerning EOs obtained from three wild species of *Satureja* from the middle of the Atlas Mountains of Morocco showed variability in the chemical composition of the EOs according to the species. *S. briquetti* essential oils showed borneol (27.64%), β-bisabolene (9.58%), α-pinene, and linalool as its main compounds, while *S. atlántica* showed piperitone oxide (27.74%%), limonene (20.57%), pulegone (16.88%), and *cis*-piperitone oxide (15.55%). Finally, the analysis of *S. alpina* presented pulegone (87.74%) as the main compound. It is known that the variation in the chemical composition of plants of the Lamiaceae family is due to environmental factors, geographic location, light intensity, soil type, altitude, and phenology, since the degree of vegetative development favors the expression of certain compounds which increase with the age of the plant [21]. This highlights the differences in the chemical composition of the EOs of *Satureja* subspecies, with great variation due to the genetic inheritance of each species and also to environmental conditions [20]. This also seems to be the case for the samples of *S. macrostema* from the Purepecha plateau.

Data regarding the antioxidant properties of *S. macrostema* EOs are shown in Table 2 and in Figure 1. Inspecting these data allows us to observe an increasing effect up to a concentration of around 50 mg/mL after which a plateau occurred in the curve of Figure 1a, in which the antioxidant effect remained almost constant, with only small increases in this effect with increasing dose of *S. macrostema* EOs. The maximum value of the DPPH antioxidant effect was 82% at a dose of 50 mg/mL of EOs. Limonene is a characteristic terpene of essential oils, and in this test, it showed a DPPH antioxidant effect of 9% with a dose of 100 mg/mL (Table 2 and Figure 1e), which was lesser than the antioxidant effect of *S. macrostema* EOs. Moreover, Trolox, a strong antioxidant often used as a reference, was the compound with the highest DPPH antioxidant effect value of 83% at a 0.1 mg/mL dose (Figure 1c). In a study on *S. macrostema* [14], the EOs demonstrated in vitro DPPH antioxidant activity of 53.11% (IC_50_) with a dose of 1 mg/mL, lower than the IC_50_ obtained in this study. Likewise, extracts of green tea and black tea with known antioxidant activity showed better antioxidant effectiveness than *S. macrostema* EOs, with DPPH greater than 80% at a lower dose of 0.5 mg/mL [22]. 

Further, essential oils from *S. macrostema* showed positive antimicrobial activity against the tested pathogenic microbial strains. The greatest effect (Table 3) on bacterial growth was with the direct application (without dilution) of 100 µL of essential oil extract, which led to 81% inhibition of *S. aureus* and 73% inhibition of *E. coli.* Bibliographic references indicate that the minimum inhibitory concentration (MIC) of EOs of *Satureja hortensis* that absolutely suppressed the visible microbial growth of several bacteria and fungi, including *S. aureus* and E. coli, was 0.2 µL EOs/mL methanol for the *S. aureus* strain [23], which is a lower dose of EOs than that used in this study. In another study on the EOs of *Satureja spicigera,* the MIC was of 1.5 mg/mL for *S. aureus* and 6 mg/mL for *E. coli* [24]. 

Essential oils are generally biologically active, with antioxidant, antibacterial, and fungicidal properties. In general, the production of essential oils and metabolites by plants is due to self-protection against the environment and against pathogens, which is why they show antibacterial activity of varying intensity depending on their origin, concentration, composition, extraction, and processing, as well as the type of microorganism [20]. 

The Lamiaceae family, to which the genus *Satureja* belongs, contains aromatic plants, many of them medicinal, with various compounds of biological interest [25]. In this study, the most active fractions in the TLC-DB test, identified as f1, f2, and f3 in Figure 2, were analyzed by gas chromatography–mass spectrometry (GC/MS). The result is shown in the Table 4, in which one can observe higher proportions of *cis*-piperitone epoxide, *trans*-piperitone epoxide, and piperitenone oxide, which were also the most abundant in the crude essential oil. Furthermore, in Table 4, it can be seen that linalool, *trans*-thujene, and *iso*-menthone are present. Since these compounds are present in the active fractions, a correlation could be established between these compounds and the antimicrobial activity of the EOs of *S. macrostema*. Regarding the most abundant compounds, it should be mentioned that piperitone (C_10_H_16_O, molecular weight: 152.23 g/mol) is a biologically active monocyclic monoterpene ketone with insecticidal activity. It is soluble in alcohol and ether with aroma of mint and camphor characteristic of eucalyptus and the genera *Piper*, *Cymbopogon*, *Andropogon*, and *Mentha.* These can be used in the production and synthesis of menthol and thymol [26,27]. In addition, piperitenone (C_10_H_14_O, molecular weight 150.22 g/mol) is a menthane monoterpenoid with antioxidant activity which is very hydrophobic and contains the strong-tasting menthol common in spearmint and rosemary [28], which generally occurs in the flowering stage and becomes pulegone or piperitone in the adult stage of the plant [29]. Likewise, piperitenone oxide (C_10_H_14_O_2_, molecular weight: 166 g/mol) is a monoterpenoid ketone that is soluble in ethanol, methanol, benzene, and petroleum [30], with cardiovascular, antimicrobial, insecticidal, and antifungal activities [31]. The biosynthetic pathway for the biogenesis of piperitone derivatives starts with the production of limonene. Then, a piperitenol compound is formed from which pulegone, menthane, and menthol are produced by one route, while piperitone, piperitone oxide, and menthol are produced by another route [17,29,32]. 

The biological activity of *S. macrotema* has been demonstrated in various tests and has been attributed to the high content of various terpenes and their synergistic action, the composition of which presents compositional variability. In general, the protection of a substrate against oxidative damage is attributed to phenolic compounds (e.g., eugenol, thymol) which are found in very low amounts in essential oils. However, since the major compounds in essential oils are monoterpenes and sesquiterpenes these compounds also contribute significantly to the antioxidant and antibacterial activity of these oils. [33]. Likewise, due to the ethnobotanical and medicinal usefulness of these plants, it is important to mention the toxicity of pulegone, since is a potent toxic compound (hepatotoxic and abortifacient) [34]; plants that contain it in high concentrations should be avoided. Piperitenone should also be investigated toxicologically because of the possible formation of the toxin *p*-cresol in plants with high contents of these compounds, as it is traditionally used in medicine [35]. Pulegone is abundant in *S. macrostema* in the early stages of development, so its use in this stage of development would not be recommended, but when it reaches a stage of greater maturity in which its concentration decreases, it can be utilized [17]. 

## 4. Materials and Methods

### 4.1. Plant Material

Maturing and wild plants (sample size *n* = 20) of *S. macrostema* were collected in 2020 in mountainous areas of the municipality of Angahuan (19°32′50″ N 102°13′34″ W, 2340 m above sea level), Uruapan Michoacán, western Mexico. The collection was carried out during the period of the most abundant plant development in this area, which corresponds to the period between September and January. The sampling area was located on the Purepecha plateau, with a height of 1700–3200 masl and cold weather almost all year round, with a temperature of 12–18 °C. The predominant vegetation consists of deciduous tropical forests, mainly including species of pines, oaks, and firs with dark or reddish soils formed from volcanic ash [36]. The leaves were detached from the plants manually and dried in a shady environment to avoid any alterations caused by solar radiation. The plant material was pulverized using a Wiley mill and sieved; the sample retained in the sieve (No. 60) was collected and stored in a refrigerator for analysis. The plants were identified at the Institute of Botany of the University of Guadalajara (IBUG), with the record number 185703.

### 4.2. Essential Oils Isolation

The essential oils were obtained by hydrodistillation based on commonly used methods [37], for which a glass system consisting of a balloon flask, condenser, heating mantle, and a continuous Clevenger trap with a 10 mL collector was necessary. Ground and washed plant material were weighed (300 g), and then 3000 mL of water were added. Heat was applied until the water was boiling, with an extraction time of 4 h. The essential oil isolated and retained in the trap, with a volume of approximately 1 mL, was separated by decantation. Residual moisture was removed from the essential oil with anhydrous sodium sulfate, stored in an amber bottle, and refrigerated at 4 °C. The necessary repetitions were made to obtain the quantity of essential oil which would be sufficient to carry out the tests.

### 4.3. DPPH and TEAC Free Radical Scavenging Assay

The following reagents were used for this test: 0.2 mM DPPH (7.89 mg DPPH in 100 mL absolute ethanol), *S. macrostema* essential oil (1 to 100 mg/mL concentrations), limonene (1 to 100 mg/mL concentrations) as reference, and Trolox (0 at 0.1 mg/mL). In addition, 0.1 M Tris-HCl (pH 7.4) was added. The measurement of activity of *S. macrostema* essential oil against DPPH radicals was carried out according to Shimamura et al. (2014) [38], with modifications. In test tubes, 200 µL of sample solution to be analyzed, 800 µL of 0.1 M Tris-HCl buffer solution (pH 7.4), and 1000 µL of 0.2 mM DPPH solution were added in consecutive order. Each preparation was mixed for 10 s. The mixtures were left to rest for 30 min in the dark. The absorbance was measured at 517 nm (Jenway 6405 Uv/Vis Spectrophotometer, Dunmow, UK), in triplicate. A mixture of 200 µL of concentrated ethanol (99.5% or 99.7%), 800 µL of 0.1 M Tris-HCl buffer solution, and 1 mL of 0.2 mM DPPH was used as a blank. The percentage of inhibition was obtained by means of the following formula:$$Inhibition\;ratio\;\left(\%\right)=\frac{A_{blank}-A_{Ssample}}{A_{blank}}\times 100$$

DPPH value was expressed in Trolox equivalents with the formula TEAC = IC_50_ Trolox (mg/mL)/IC50 sample (mg/mL). Trolox (6-hydroxy-2,5,7,8-tetramethylchroman-2-carboxylic acid) is a vitamin E-derived antioxidant with free radical activity. In many studies, the antioxidant activity of the samples against free radicals (DPPH, ABTC) is compared against Trolox as a standard antioxidant by means of the corresponding ratio (Trolox inhibition/compound inhibition), known as TEAC [39].

### 4.4. Antimicrobial Activity In Vitro

Antimicrobial tests were performed using the broth microdilution assay [40]. Essential oil serial dilutions of 1/10, 1/100, 1/500, and 1/1000 (*v***/***v*) were made. In addition, bacterial suspensions of *Escherichia coli* (ATCC^®^ 8739) and *Staphylococcus aureus* (ATCC^®^ 6538p), previously cultured overnight at 30 °C in Mueller–Hinton broth (MHB), were adjusted to densities of 0.5 according to the turbidity scale of McFarland (1.5 × 10^8^ CFU/mL). Then, 100 µL of the extracts and 100 µL of the prepared bacterial suspensions were deposited in a 96-well microtiter plate (Costar TM, made of polystyrene). The inoculated plates were incubated at 37 ± 0.1 °C at 24 h with continuous agitation. The samples with bacteria and without extract were considered as controls. The antimicrobial activity (inhibition of growth and multiplication bacterial) was determined by measuring the absorbance at 620 nm using a spectrophotometer (Jenway 6405 Uv/Vis Spectrophotometer, UK). These measurements were performed in triplicate.

### 4.5. Thin Layer Chromatography–Direct Bioautography (TLC-DB)

This test was developed according to the method described by Jesionek et al. (2015) [41], with modifications. A suspension of microorganisms of *Escherichia coli* (ATCC^®^ 8739) and *Staphylococcus aureus* (ATCC^®^ 6538p) was prepared in Mueller–Hinton agar (MHA). Subsequently, they were inoculated in 100 mL of nutrient broth (0.5% agar) at 37 °C for 48 h. The concentration was adjusted by dilution with nutrient broth and absorption from 0.4 to 600 nm, corresponding to 4 × 10^7^ UFC/mL. TLC plates were prepared with fluorescence factor (60 Merck^®^, Darmstadt, Germany, 10 × 10 cm), conditioning them at 45 °C for 12 h. For the development of the plates, approximately 10 µL of a methanol solution of the essential oils (30 µL/mL) of *S. macrostema* was deposited at the base of the plates, and solutions of *M. Pugana* and *T. vulgaris* were used for comparison. Elution was performed with toluene/ethyl acetate/petroleum ether (93:7:20) *v*/*v*/*v*. The bands eluted by fluorescence were observed at 254 nm. The test was carried out in triplicate. The plates were placed in a chamber at 25 °C for 90 min. They were immersed for 10 s in 50 mL of the bacterial suspension, removed, and dried by air flow, then placed into a steam chamber at 37 °C for 24 h in an incubator (Binder BD115, Camarillo, CA, USA). Subsequently, the plates were immersed for 60 s in aqueous tetrazolium salt MTT (3-(4,5-dimethylthiazol-2-yl)-2,5-diphenyl tetrazolium bromide) solution (Sigma Aldrich, Burlington, MA, USA, 0.6 mg/mL) with 0.1% Triton X-100, then removed and incubated for 1 h in a steam chamber at 37 °C. They were then immersed in 50 mL of ethanol (70% *v*/*v*) for 10 s and dried with air. The occurrence of light bands or spots indicated antibacterial activity in contrast to the bluish background. The distances were measured to calculate the Rf. The strongest light points indicated the fractions with the highest antimicrobial intensity levels. The fractions with coincident Rf were located, the silica was carefully scraped off with a spatula, and the contents were stored in an amber vial. Subsequently, the silica with the active fraction was diluted in 2 mL of dichloromethane (Sigma-Aldrich Merck subsidiary Mexico, Naucalpan de Juárez, Mexico), then filtered on PDVF membranes with 0.45 µm porosity and injected into the GC-MS equipment (Bruker Daltonics 436 SCION^®^, Bremen, Germany). The same parameters as those indicated in the chemical characterization of the essential oils were used. 

### 4.6. Gas Chromatography-Mass Spectrometry

The *S. macrostema* EOs were analyzed by GC-MS according to the method used by Scalvenzi et al. 2017 [42]. A gas chromatograph (Bruker Daltonics 436 SCION^®^, Bremen, Germany) coupled to a triple quadrupole mass detector (model TQ-EVOQ) was used. A Bruker BR-5MS GC column (30 × 0.25 mm × 0.25 µm, Bremen, Germany) was also used. The maximum analysis temperature was 350 °C. The temperature program was as follows: 50 °C to 100 °C at 1 °C/min; 100 °C to 250 °C at 5 °C/min, and maintenance of this temperature for 10 min for a total analysis time of 90 min. The mass spectrometer conditions were ionization energy, 70 eV; emission current, 10 mAmp; scan rate, 1 scan/s; mass range, *m*/*z* 35–400 Da; trap temperature, 220 °C; and transfer line temperature, 260 °C. The carrier gas was helium (99.999%), with a flow 1 mL/min and a split radius of 1:50. The LIR values were calculated according to the formula proposed by Van Den Dool and Kratz, 1963 [43]. These were applied to programs with thermal gradients, using C_8_ to C_20_ alkanes with their respective retention times as standards, then compared against the retention rates reported in the literature [13]. The components were identified based on IR and by comparison with the mass spectra essential oil component libraries [12,13]. Bruker’s MS-Workstation software integrated with NIST-MS-Search Ver. 8.2.1 was used for this analysis.

## 5. Conclusions

The main biological properties of *S. macrostema* were evaluated due to its ethnomedicinal use to combat various diseases. A high percentage of volatile terpenes was found, most of them being oxygenated monoterpenes, with a high proportion of piperitone oxides and epoxides as well as linalool, pulegone, menthone, and limonene, among other common components in essential oils of aromatic plants with medicinal properties. The chemical composition of *S. macrostema* EOs showed variation in the type and quantity of compounds due to edaphoclimatic and phenological factors of the growing area. Likewise, the biological activity of these compounds was attributed more to synergistic action than to the individualized activity of the compounds in in vitro investigations. The evaluation of their biological properties is important for the characterization and knowledge of these biotic resources and may even guide the direct measurement of their effects on various diseases, including chronic degenerative diseases (e.g., diabetes, cancer, hypertension), either in cell lines or in controlled clinical studies. 

## Figures and Tables

**Figure 1 molecules-28-04719-f001:**
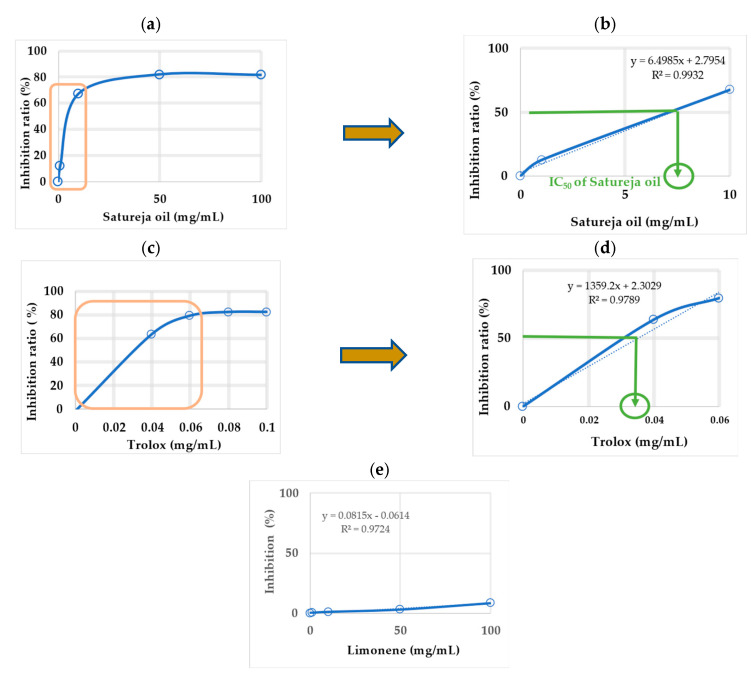
Graphs of the antioxidant effects at different concentrations of antioxidant compounds and IC_50_ determination of the DPPH antioxidant assay. (**a**,**c**): plots of the inhibition ratio (y) vs. the sample concentration (x) of S. macrostema EOs and Trolox; (**b**,**d**): plots of the zone containing 50% free radical scavenging (IC_50_) of *S. macrostema* essential oils and Trolox; (**e**) antioxidant effect of limonene (in this case, IC_50_ was obtained by extrapolation).

**Figure 2 molecules-28-04719-f002:**
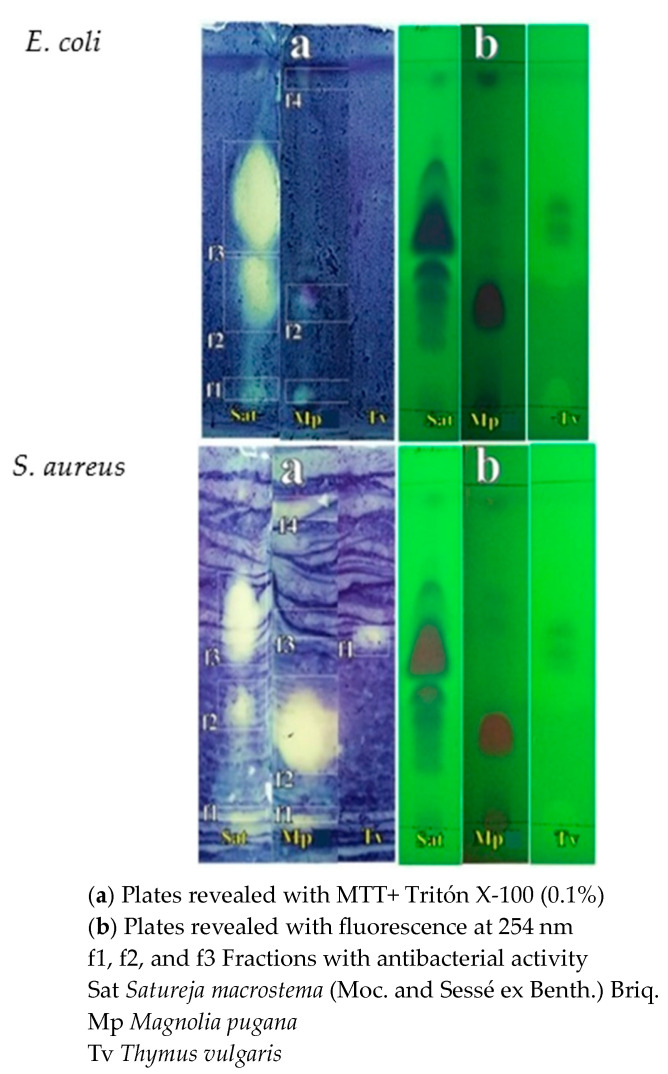
TLC-DB of the essential oils from *Satureja macrostema* (Moc. and Sessé ex Benth.) Briq. And their activity against *E. coli* and *S. aureus*, compared with essential oils from the leaves of *Magnolia pugana* (Mp) and *Thymus vulgaris* (Tv). White and gray spots show fractions with antimicrobial activity.

**Table 1 molecules-28-04719-t001:** Chemical composition of *Satureja macrostema* (Moc. and Sessé ex Benth.) Briq. essential oil.

No.	Compound	LRI (Calculated)	LRI (Literature)	CAS #	Retention Time (Min)	% Relative Area	S.D.
1	2E-hexenal	845	846	6728-26-3	4.2	0.08	±0.0
2	α-pinene	928	932	99-83-2	6.3	0.1	±0.0
3	β-thujene	967	968	28634-89-1	7.8	1	±0.0
4	β-pinene	972	974	127-91-3	8.0	0.3	±1.0
5	3-octanol	994	988	589-98-0	8.8	0.3	±0.1
6	limonene	1021	1024	138-86-3	10.6	0.4	±0.1
7	linalool	1094	1095	78-70-6	16.1	5	±1.0
8	1-octen-3-yl-acetate	1105	1110	2442-10-6	17.1	0.2	±0.0
9	3-octanol, acetate	1116	1120	4864-61-3	18.4	6	±0.2
10	menthone	1145	1148	89-80-5	21.9	3	±0.0
11	*iso*-menthone	1154	1150	491-07-6	22.9	0.2	±0.1
12	*neoiso*-menthol	1160	1184	491-02-1	23.6	0.1	±0.1
13	pulegone	1226	1233	89-82-7	31.7	3	±0.4
14	*cis*-piperitone epoxide	1240	1250	4713-37-5	33.4	22	±1.4
15	*trans*-piperitone epoxide	1243	1252	57130-28-6	33.7	46	±4.2
16	acetic acid, 2-phenylethyl ester	1246	1254	103-45-7	34.1	1	±0.0
17	piperitenone oxide	1360	1366	35178-55-3	47	11	±1.1
18	caryophyllene	1404	1417	87-44-5	50.5	1	±0.1
19	bicyclogermacrene	1486	1500	24703-35-3	54.0	0.2	±0.0
20	spathulenol	1568	1577	6750-60-3	56.8	0.3	±0.1
21	cyclocolorenone	1747	1759	489-45-2	61.5	1	±0.5
	Not identified					0.7	
	Oxygenated Hydrocarbons					0.3	
	Monoterpenes					1	
	Oxygenated monoterpenes					96	
	Sesquiterpenes					1	
	Oxygenated sesquiterpenes					1	
	Total					100	

Note: Components are listed in order of elution, and nomenclature is in accordance with NIST (National Institute of Standards and Technology). LRI (Calculated): The calculated linear retention index was obtained by comparison to an alkane C_7_–C_20_ standard mixture analyzed in the same conditions as the samples on a non-polar column. LRI (Literature): linear retention index in literature. CAS #: registry number assigned by the Chemical Abstracts Service. S.D.: standard deviation.

**Table 2 molecules-28-04719-t002:** In vitro antioxidant activity of essential oils of *Satureja macrostema*. (Moc. and Sessé ex Benth.) Briq.

mg/mL	$\%\mathrm{DPPH},\;\left(\bar{x}\;\pm SD\right)$		
	Essential Oil	Limonene	Trolox
0.00	nd	nd	nd
0.04	nd	nd	51 ± 2.3
0.06	nd	nd	79 ± 1.5
0.08	nd	nd	82 ± 0.2
0.1	nd	0.3 ± 0.16	83 ± 0.1
1	12 ± 0.4	0.2 ± 0.1	nr
10	67 ± 1.7	1 ± 0.8	nr
50	82 ± 0.2	3 ± 1.1	nr
100	82 ± 0.2	9 ± 0.9	nr

nd = not detectable; nr = not carried out.

**Table 3 molecules-28-04719-t003:** In vitro bacterial growth inhibition of *Satureja macrostema* (Moc. and Sessé ex Benth.) Briq. essential oils over *E. coli* and *S. aureus*.

Dilution, *v*/*v*, %	% Inhibition, $\left(\bar{x}\;\pm SD\right)$	
	*E. coli*	*S. aureus*
Direct (100%)	73 ± 1.6	81 ± 1.7
1:10 (10%)	51 ± 2.2	46 ± 3.2
1:100 (1%)	31 ± 1.4	28 ± 2.5
1:500 (0.5%)	6 ± 0.9	20 ± 2.9
1:1000 (0.1%)	nd	6 ± 2.1

nd = not detectable.

**Table 4 molecules-28-04719-t004:** Active antibacterial fractions in the essential oils of *Satureja macrostema.* (Moc. And Sessé ex Benth.) Briq. Obtained via the TLC-DB analysis. F1, f2, and f3 are the zones of the thin layer in the TLC-DB test with bacterial activity, and Rf is the retention factor of each of these active zones. Compounds found by GC-MS from each of these areas are included.

*S. macrostema* Essential Oils Fractions with Antimicrobial Properties			
	f1	f2	f3
Retention factor (*Rf*) values			
*E. coli* *S. aureus*	0.02 0.02	0.31 0.35	0.62 * 0.61 *
Detected compounds			
	linalool (>) caryophyllene *trans*-thujene piperitenone spathulenol	pulegone piperitone oxide (>)	*cis*-piperitone epoxide *trans*-piperitone epoxide (>) *iso*-menthone 2-phenil ethyl acetate

* Antibacterial fraction with major inhibition; (>) compound with the highest presence inside the antibacterial fraction.

## Data Availability

Data presented in this study are available upon request from the corresponding author.

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
