# Peer review of "Chemical Composition, Antioxidant Properties, and Antibacterial Activity of Essential Oils of Satureja macrostema (Moc. and Sessé ex Benth.) Briq"

_molecules, 2023, doi:10.3390/molecules28124719_

Round 1

Reviewer 1 Report (New Reviewer)

The manuscript “Chemical composition, Antioxidant Properties and Antibacterial Activity of Essential Oils of Satureja macrostema (Moc. and Sessé ex Benth.) Briq.” deals with the extraction of essential oil from Satureja macrostema leaves and its chemical and antimicrobial characterization. The work is interesting and well organized. However, it requires some revisions before the publication, as follows:

- Introduction. The state of the art concerning the extraction of essential oil and active compounds from vegetable matter has to be enlarged; moreover, a short description of the innovative extraction methods can be added to underline cons/pros. For this purpose, see for instance these recent works: Baldino et al., Concentration of Ruta graveolens active compounds using SC-CO2 extraction coupled with fractional separation, Journal of Supercritical Fluids, 2018, 131, pp. 82–86; Laurintino et al., Evaluation of the biological activity and chemical profile of supercritical and subcritical extracts of Bursera graveolens from northern Peru, Journal of Supercritical Fluids, 2023, 198, 105934; etc…

- R&D. Improve the description and the comparison of the results obtained with the previous literature in order to highlight the relevance of the present findings.

Minor editing of English language required.

Author Response

Dear Reviewer, the authors of this manuscript reiterate our gratitude for your time invested in reviewing and making comments to improve our work, as well as the result of the evaluation made. Please see the attachment.

Kind regards

Sincerely

The manuscript authors

Reviewer 2 Report (New Reviewer)

Chemical composition, Antioxidant Properties and Antibacterial Activity of Essential Oils of Satureja macrostema (Moc. and 3 Sessé ex Benth.) Briq.

ABSTRACT

Line 15-20:  The author should recast the sentence the essential oils of its leaves were obtained, and  the chemical composition was evaluated by means of Gas Chromatography-Mass Spectroscopy  (GC-MS) as well as the 2,2- diphenyl-1-picrylhydrazyl (DPPH) and Trolox equivalent antioxi- 17 dant capacity (TEAC) antioxidant method, in vitro antibacterial using broth microdilution assay and 18 by means of Thin Layer Chromatography-Direct bioautography TLC-DB to identify antibacterial 19 compounds active against Escherichia coli and Staphylococcus aureus.

The sentence can be written as the essential oils were obtained from leaves Satureja macrostema, and the chemical composition was evaluated by Gas Chromatography-Mass Spectroscopy (GC-MS). The antioxidant potential of the oil was assayed by 2,2- diphenyl-1-picrylhydrazyl (DPPH) and Trolox equivalent antioxidant capacity (TEAC) and the in vitro antibacterial activity was determined using broth microdilution assay and Thin Layer Chromatography-Direct bioautography TLC-DB.

Line 22: This statement should be represented in a more comprehensive format “The biologicaltests obtained were DPPH = 82%, Half Maximal inhibitory concentration (IC50) = 7 mg/mL, TEAC 23 = 0.005, Inhibition of 73 % on E. coli and 81% on S. aureus with 100 µL of undiluted crude oil”.

RESULTS

Line 67-71, 79-83: The author should write only the title of the table on the table and every other word Like component listed in order of elution etc should be in the introduction section of table 1 where a total of 21 compound was written (heading 2.1).

Line 88-91, 94-103: The author should write only the title of the table on the table and every other word should be in the introduction section of table 2 where DPPH and TEAC free radical scavenging assay was written (heading 2.2).

Line 94-103:  This should be under the heading 2.2 and the then infer to the figure 1 in the description for better clarification of the result.

Line135-136 ,139-142: The author should write only the title of the table on the table and the different concentrations used should be stated under heading 2.3, introduction to the table 3.

Line 139-142: Author should write this statement under the heading 2.3 and refer to table 3.

Line 147: EOs from leaves of Magnolia pugan (Mp) and Thymus vulgaris (Tv) were taken as a 147 comparison. This statement should be recast.

Line 161- 164, 175-178: The statement should be written under heading 2.4 introduction section of figure 2 and table 4.

DISCUSSION

Lines 184- 190, 229- 239: Why are you repeating result in the discussion section.

The discuss section is to compare the finding of your work with the findings of previous authors and linked this to any justification for the findings.

MATERIALS AND METHODS

Line 301: what does author mean by n=20?

Line 304: September and January should be rewrite as September and January.

Line 408: Showed was duplicated and one should be deleted.

CONCLUSION

Line 415-416: I don’t think this statement “Also is important stablish the cytotoxic effect of some species of this genus containing high content of pulegone or Piperitenone, potentially toxic com pounds”. Is necessary in this section

The authors need to improve the quality of English used in this article.

Author Response

Dear Reviewer, we appreciate your accurate comments that will allow a better 
version of our manuscript. We hope we have responded to your comments. Thanks for your time and effort. Please see the attachment.

Kind regards. 

Sincerely

Manuscript authors

This manuscript is a resubmission of an earlier submission. The following is a list of the peer review reports and author responses from that submission.

Round 1

Reviewer 1 Report

1. As described in 4.2, about 1 mL oil was obtained from each 300g plant material, how many repeats of EO isolation were conduted to surpport this study? For example, in the antimicrobial tests, a dose of 100 µL was applied.

2. Abstract: with direct dose of 100 µL, how about use the expression of x µL/mL of the EO used in antimicrobial or antioxidant evaluation? 

3. Table 2: Oil ?

Author Response

-As described in 4.2, about 1 mL oil was obtained from each 300g plant material, how many repeats of EO isolation were conduted to surpport this study? For example, in the antimicrobial tests, a dose of 100 µL was applied.

R. In the methodology section it is specified that for the hydrodistillation of the essential oil this quantity of 300g of leaves and 3000 mL of water was used (thanks to your observation this quantity was corrected in the text since it presented an error, paragraph 314), due to to the capacity of the 5-L flask used in conjunction with the Clavenger trap. Several repetitions were made, approximately 10 times until the leaves of the sampled material were exhausted and there was enough essential oil to carry out the tests. A paragraph was added in this regard (paragraph 310-311).

-Abstract: with direct dose of 100 µL, how about use the expression of x µL/mL of the EO used in antimicrobial or antioxidant evaluation?

R. This data corresponds to the best result obtained in the antimicrobial test and refers to the fact that a dose of 100 undiluted crude essential oil was applied. A statement is added to better specify this (line 25). Likewise, in this antibacterial assay, various concentrations of essential oils were used, making various dilutions of the crude essential oil obtained, expressed as a v/v dilution and in percentage (%), concentration units transformable to the concentration µL/ mL This way of expressing concentrations is used interchangeably in reports.

-Table 2: Oil ?

R. Oil refers to essential oils obtained from S. macrostema. To avoid confusion, we have made a revision to add the word “essential” where only “oil” was specified, throughout the document.

Dear reviewer, thank you for your time and effort.

Reviewer 2 Report

In this manuscript, the authors determined chemical composition of S. macrostema on the Purepecha plateau and the effect on E. coli and S. aureus, evaluating the chemical compounds responsible for this microbicidal activity. The article is relatively simple and does not delve into the reasons. There are some points that should be revised. My major comments are in the following: 

Abstract

This part needs to list the important results and also clarify the purpose and significance of the experiment.

The keywords in this section need to be modified.

Results

2.2 Line 80: The relationship between x and y marked in this sentence does not match the note of figure 1.

Discussion

Paragraph content lacks logic and needs to be reordered and reworded.

Please add your own analysis and discussion while comparing the results with other literature, otherwise the repetition is too high.

Conclusions

This section is too general and could highlight the key results of the experiment, while adding the shortcomings of the current experiment and future research directions. 

Author Response

In this manuscript, the authors determined chemical composition of S. macrostema on the Purepecha plateau and the effect on E. coli and S. aureus, evaluating the chemical compounds responsible for this microbicidal activity. The article is relatively simple and does not delve into the reasons. There are some points that should be revised. My major comments are in the following:

Abstract

-This part needs to list the important results and also clarify the purpose and significance of the experiment.

R. The abstract was modified in this sense. The results of the analysis of essential oils and the most important conclusions were included. We consider that the significance and purpose of the study were already well specified.

-The keywords in this section need to be modified.

R. One of the keywords was modified, removing the keyword "Chemical characterization" adding "Piperitone derivatives" instead. The rest of the keywords we believe adequately represent the content and purpose of the manuscript and will be useful in the search for potential readers. Thanks for your observation.

Results

-2.2 Line 80: The relationship between x and y marked in this sentence does not match the note of figure 1.

R. A paragraph (125-126) is added in the description of figure 1 to match line 80 (line 92 in the edited manuscript).

Discussion

Paragraph content lacks logic and needs to be reordered and reworded. Please add your own analysis and discussion while comparing the results with other literature, otherwise the repetition is too high.

R. We appreciate your comment. The "discussion" section is very important in the manuscript, so it is the longest and the one we put a lot of effort into. We sincerely believe that there is no unnecessary repetition of data, analysis, and commentary of the same. What it does contain is a comparison with several studies of the same plant under different conditions and different results, especially in the chemical composition of essential oils, in each case we issue our own analyses, comments and observations. This which we believe that not only favors and contributes on the content of the manuscript but is also something necessary. Thus, sincerely we believe that it does not need to be rearranged and rewritten. We respectfully hope that you will consider these comments.

Conclusions

This section is too general and could highlight the key results of the experiment, while adding the shortcomings of the current experiment and future research directions.

R. Agree. A paragraph was added (paragraph 390-398) commenting on the most relevant results, as well as a reflection on the trend of future research on compounds obtained from aromatic plants with medicinal properties (paragraph 402-406).

 Dear reviewer, thank you for your time and effort.

Reviewer 3 Report

The paper submitted for review: Chemical composition and Antibacterial Activity of Essential Oils of Satureja macrostema (Moc. and Sessé ex Benth.) Briq. is an analysis of the chemical composition of the essential oil obtained from the leaves of Satureja macrostema and the assessment of antioxidant and antimicrobial activity against two selected pathogens.

The topic chosen by the Authors is interesting, but the manuscript in its current form does not meet the criteria for publication in the journal: Molecules.

Title: does not exhaust the issues discussed in the manuscript.

Abstract: contains unexplained abbreviations, conclusions from the results are not shown.

Keywords: please think about whether the keywords indicated here properly characterize the conducted research and will be able to help people interested in this topic find this particular job in the future.

Scientific Background: This chapter is correct.

Results: Any abbreviation used must be entered first to make the text unambiguous for readers. Descriptions of tables and figures must comply with the principle of self-description and be unambiguous even after the table or figure has been removed from the manuscript. Therefore, I am asking to correct the descriptions of the tables and figures (first of all, please indicate the full name of the plant from which the essential oil was extracted).

Please note the distinction between: oil and essential oil. This difference is very important and means completely different plant materials with a completely different chemical composition. Therefore, I would like to ask to consistently use the term: essential oil.

I also ask to consistently write terms in Latin in italics (currently there is chaos here). Also, please write the names of the phytochemical components of the essential oil in lower case (here there is chaos too).

If the numeral is at the beginning of a sentence, it should be written as a word, not a number.

I don't understand the sentence on line 123.

Discussion: Currently, this chapter is very poor. The authors mix the data on the essential oil obtained from the leaves and from the seeds.

The discussion on antioxidant and antibacterial activity was combined into one paragraph. It surprises. The reader has the impression that antimicrobial activity has been treated as less important in general.

Material and method: methodological descriptions lack many elements that would allow reconstructing such research. There is no data on the equipment used, and even if the name of the equipment is indicated, the manufacturer and country of production are not given. Please provide detailed methodological descriptions. Lines 322 and 351: please remove the brackets from the publication year of the cited publications.

Conclusion: this chapter is too general. Please indicate the most important results. Please indicate the application conclusions and the direction of further research, if any.

References: The chapter needs a complete rearrangement

Author Response

REVIEWER 3

The paper submitted for review: Chemical composition and Antibacterial Activity of Essential Oils of Satureja macrostema (Moc. and Sessé ex Benth.) Briq. is an analysis of the chemical composition of the essential oil obtained from the leaves of Satureja macrostema and the assessment of antioxidant and antimicrobial activity against two selected pathogens.

The topic chosen by the Authors is interesting, but the manuscript in its current form does not meet the criteria for publication in the journal: Molecules.

-Title: does not exhaust the issues discussed in the manuscript..

R. The sentence “Antioxidant Properties” was added to the title, with which we consider that the title fully expresses the content of the manuscript.

-Abstract: contains unexplained abbreviations, conclusions from the results are not shown.

R. The meaning of the abbreviations GC-MS, DPPH, TEAC, TLC-DB were included. A new text was added with data on the chemical composition of the essential oil (paragraphs 20-23) and on the main conclusions (paragraphs 26-28).

-Keywords: please think about whether the keywords indicated here properly characterize the conducted research and will be able to help people interested in this topic find this particular job in the future.

R. One of the keywords was modified, removing the keyword "Chemical characterization" adding "Piperitone derivatives" instead. The rest of the keywords we believe adequately represent the content and purpose of the manuscript and will be useful in the search for potential readers. Thanks for your observation.

-Scientific Background: This chapter is correct.

R. Thanks for your generosity.

-Results: Any abbreviation used must be entered first to make the text unambiguous for readers. Descriptions of tables and figures must comply with the principle of self-description and be unambiguous even after the table or figure has been removed from the manuscript. Therefore, I am asking to correct the descriptions of the tables and figures (first of all, please indicate the full name of the plant from which the essential oil was extracted).

R. A greater description of Table 1, Table 2, Table 3 and Figure 2 was made, which can be appreciated by the text highlighted in green. Likewise, the full name of Satureja macrostema was added.

-Please note the distinction between: oil and essential oil. This difference is very important and means completely different plant materials with a completely different chemical composition. Therefore, I would like to ask to consistently use the term: essential oil.

-Agree, thank you, the term "essential oils" was standardized in the following lines: line 53, line 83, Table 2, line 96, line 99, line 128, line 138, line 188, line 198, line 201, line 222, line 228, line 234, line 236, line 241, line 253, line 281, line 307, line 316.

-I also ask to consistently write terms in Latin in italics (currently there is chaos here). Also, please write the names of the phytochemical components of the essential oil in lower case (here there is chaos too).

R. A revisión of the manuscript was made correcting:

a) the latin terms in italics: word in vitro, line 230.

b) phytochemical compounds were also expressed in lower case: compounds from column 1 of table 1; limonene and trolox from table 2; line 128, line 183; line 226; line 228; line 315; line 326; line 330; line 360; line 368

-If the numeral is at the beginning of a sentence, it should be written as a word, not a number.

R. OK. A review of the document was made without detecting a situation of this type.

-I don't understand the sentence on line 123.

R. It refers to the interpretation of the data shown in table 3, where it is mentioned that the greatest effect was when the essential oil was applied undiluted and that the least resistant microorganism was S. aureus compared to E. coli. This paragraph was written again, more explicitly (paragraph 137-140).

-Discussion: Currently, this chapter is very poor. The authors mix the data on the essential oil obtained from the leaves and from the sedes.

R. For this important section, a comparison of the results is made against studies of essential oils of the same species, S. macrostema first, and against other species of satureja. Likewise, paragraph 169-173 of reference 11 was modified to specify that the study was carried out on essential oils from the aerial part (leaves and stems) of plants obtained by micropropagation from seeds collected in an experimental plantation (paragraph 188 -190).

-The discussion on antioxidant and antibacterial activity was combined into one paragraph. It surprises. The reader has the impression that antimicrobial activity has been treated as less important in general.

R. Agree, thanks. These two paragraphs were separated to discuss them separately, assigning the corresponding importance to these properties separately.

-Material and method: methodological descriptions lack many elements that would allow reconstructing such research. There is no data on the equipment used, and even if the name of the equipment is indicated, the manufacturer and country of production are not given. Please provide detailed methodological descriptions. Lines 322 and 351: please remove the brackets from the publication year of the cited publications.

R. In the methodological description, an attempt is made to make a detailed description of the experimentation, so that it can be reproduced in similar investigations. In relation to the data of the equipment used, the description of the spectrophotometer used to measure the antioxidant activity (line 321), the antimicrobial activity (line 343), the country (United States) manufacturer of the Binder incubator for the TLC-DB test was added (Line 359), the country (Germany) manufacturer of the Gas Chromatograph used to identify the constituent compounds of essential oils (line 370 and line 374, the country manufacturer of the Bruker BR-5MS GC column (Line 383). Also were removed the indicated brackets of references (Line 345 and Line 374).

-Conclusion: this chapter is too general. Please indicate the most important results. Please indicate the application conclusions and the direction of further research, if any.

R. In conclusions, comments were added in the form of a summary on the most relevant results found, especially the chemical composition of the essential oil obtained and its biological properties (paragraphs 390-398). A paragraph (402-406) was also added suggesting future studies of aromatic plants with medicinal properties, as is the case.

-References: The chapter needs a complete rearrangement

R. A review of the format of the references was made, following the indications of the author's guide for its uniformity.

Dear reviewer, thank you for your time and effort.

Round 2

Reviewer 2 Report

I have read carefully the authors' revisions, and I think the revised manuscript may be accepted  for publication now.

Reviewer 3 Report

The manuscript has not been improved sufficiently. This applies primarily to the chapters: Discussion, Conclusions and References. 

The discussion is poor, the corrections made after the first round of reviews are two lines of text and the correction of mistakes in the form of calling essential oils as: oils. This is surprising, especially since two out of three reviewers pointed out major shortcomings in this chapter.

Conclusions are still incorrect. Instead of making corrections, the chapter was unnecessarily extended.

Although the authors wrote in their answers that the references were corrected - this chapter is incorrectly prepared.